# Decentralising chronic disease management in sub-Saharan Africa: a protocol for the qualitative process evaluation of community-based integrated management of HIV, diabetes and hypertension in Tanzania and Uganda

Marie-Claire Van Hout [1], Mathias Akugizibwe,[2] Elizabeth Henry Shayo [3], Moreen Namulundu,[2] Francis Xavier Kasujja [2], Ivan Namakoola,[2] Josephine Birungi,[2] Joseph Okebe,[4] Jamie Murdoch,[5] Sayoki Godfrey Mfinanga [6], Shabbar Jaffar[4]

For numbered affiliations see end of article.

**Correspondence to**
Professor Marie-Claire Van Hout;
M.C.VanHout@ljmu.ac.uk

## ABSTRACT

**Introduction** Sub-Saharan Africa continues to experience a syndemic of HIV and non-communicable diseases (NCDs). Vertical (stand-alone) HIV programming has provided high-quality care in the region, with almost 80% of people living with HIV in regular care and 90% virally suppressed. While integrated health education and concurrent management of HIV, hypertension and diabetes are being scaled up in clinics, innovative, more efficient and cost-effective interventions that include decentralisation into the community are required to respond to the increased burden of comorbid HIV/NCD disease.

**Methods and analysis** This protocol describes procedures for a process evaluation running concurrently with a pragmatic cluster-randomised trial (INTE-COMM) in Tanzania and Uganda that will compare community-based integrated care (HIV, diabetes and hypertension) with standard facility-based integrated care. The INTE-COMM intervention will manage multiple conditions (HIV, hypertension and diabetes) in the community via health monitoring and adherence/lifestyle advice (medicine, diet and exercise) provided by community nurses and trained lay workers, as well as the devolvement of NCD drug dispensing to the community level. Based on Bronfenbrenner's ecological systems theory, the process evaluation will use qualitative methods to investigate sociostructural factors shaping care delivery and outcomes in up to 10 standard care facilities and/or intervention community sites with linked healthcare facilities. Multistakeholder interviews (patients, community health workers and volunteers, healthcare providers, policymakers, clinical researchers and international and non-governmental organisations), focus group discussions (community leaders and members) and non-participant observations (community meetings and drug dispensing)

## STRENGTHS AND LIMITATIONS OF THIS STUDY

⇒ The INTE-COMM trial responds to the continued need to tackle the increased burden of non-communicable diseases (NCDs) in sub-Saharan Africa and decentralise healthcare systems into the community to manage comorbidities and multimorbidities (diabetes and hypertension) with HIV.

⇒ A pragmatic cluster-randomised trial will compare community-integrated care for HIV, diabetes and hypertension with standard facility-based integrated care for the first time in Tanzania and Uganda.

⇒ The INTE-COMM trial is based on a partnership between African and UK researchers, working closely with policymakers, service user representatives, healthcare providers and community members to provide integrated care for people living with HIV, diabetes or hypertension in the community.

⇒ The process evaluation of INTE-COMM employs qualitative interviews and observational methods at 4–10 sites to explore and document the delivery of integration care in the community setting, with a view to understanding the role of individual, community, familial, social and structural factors shaping implementation and outcomes.

⇒ Foreseen limitations of the process evaluation may include community site characteristics, lack of service user and community buy-in due to perceptions that community care is inferior to facility-based integrated care, stigma, lack of trust in community drug dispensing issues, service user drop-out, social desirability bias and various other contextual mitigating factors.

will explore implementation from diverse perspectives at three timepoints in the trial implementation. Iterative sampling and analysis, moving between data collection points and data analysis to test emerging theories, will continue until saturation is reached. This process of analytic reflexivity and triangulation across methods and sources will provide findings to explain the main trial findings and offer clear directions for future efforts to sustain and scale up community-integrated care for HIV, diabetes and hypertension.

**Ethics and dissemination** The protocol has been approved by the University College of London (UK), the London School of Hygiene and Tropical Medicine Ethics Committee (UK), the Uganda National Council for Science and Technology and the Uganda Virus Research Institute Research and Ethics Committee (Uganda) and the Medical Research Coordinating Committee of the National Institute for Medical Research (Tanzania). The University College of London is the trial sponsor. Dissemination of findings will be done through journal publications and stakeholder meetings (with study participants, healthcare providers, policymakers and other stakeholders), local and international conferences, policy briefs, peer-reviewed journal articles and publications.

**Trial registration number** ISRCTN15319595.

## INTRODUCTION

The global burden of non-communicable diseases (NCDs) is recognised in the Sustainable Development Agenda 2023 and the sustainable development goals (SDG) as a substantial challenge for sustainable development.[1] Most recent global NCD data from the 2019 Global Burden of Disease Study has reported that NCDs kill 41 million people annually (equivalent to 74% of all deaths globally). This study has also documented a concerning global increase in NCD deaths over time, with the percentage of deaths due to NCDs increasing from 60% in 1999 to 74% in 2019 (approximately 15% increase over two decades).[2] The majority of NCD deaths are due to cardiovascular diseases (estimated at 17.9 million annually), followed by cancer (9.3 million), chronic respiratory diseases (4.1 million) and diabetes (2 million, which includes kidney disease deaths due to diabetes).[1] 17 million people are reported to die from an NCD before the age of 70.[1 2]

NCDs present enormous challenges to progress towards attaining the targets set in the Sustainable Development Agenda 2030. The SDG target (3.4) aims to reduce premature mortality from NCDs by one-third through prevention and treatment by 2030.[1] Prevention, screening, treatment and care of NCDs (including palliative care) are key components of the NCD response.[1] Various modifiable behavioural risk factors increase the risk of NCD-related mortality (eg, tobacco and alcohol use, excess salt intake, unhealthy diets and physical inactivity). Metabolic risk factors, such as obesity, hyperglycaemia, hypolipidaemia and hypertension, increase the risk of NCD disease.[1 2] Other environmental factors, such as air pollution, cause NCDs such as lung cancer, chronic obstructive pulmonary disease, ischaemic heart disease and stroke.[1 2]

Low-income and middle-income countries (LMICs) are, however, disproportionately impacted by NCD morbidity and mortality.[3–6] Poverty reduction initiatives are hindered by rising HIV/NCD multimorbidities due to increased household costs related to healthcare and higher mortality and morbidity among the vulnerable and socially disadvantaged.[1] Of all NCD deaths, 77% are in LMICs, where 86% of NCD deaths before the age of 70 occur.[1 2] The Global Alliance of Chronic Diseases also observes that of the more than 15 million people aged 30–69 who die annually due to NCDs, 85% of these premature deaths occur in LMICs. Hypertension and diabetes cause the most NCD-related mortality and morbidity in LMICs.[7 8] Women living in LMICs are also disproportionately affected by the triple burden of reproductive and maternal health conditions, NCDs and HIV.[4–6]

### Sub-Saharan Africa (SSA)

The SSA region continues to experience a syndemic of HIV and NCDs.[1 9–14] The region accounts for 55% of the total 38 million people living with HIV (PLHIV) globally.[15] In addition to the rise in patients in regular HIV care (almost 80%) and virally suppressed (90%),[16] NCDs among PLHIV in SSA are also increasing. This is due to various HIV-related reasons (HIV infection itself, antiretroviral treatment (ART)) and ageing-related chronic comorbidities (diabetes, hypertension, cancers and metabolic disorders).[17–20] Key relevant socioeconomic factors fueling the rise of NCDs in the region are increased urbanisation, poor lifestyles and poverty, NCD-related healthcare costs (eg, drugs) and vulnerable and poor populations substantially impacted by chronic illness.[9 11 12 21]

Long-term management of diabetes and hypertension among PLHIV remains a major health system challenge in SSA,[22 23] where rising HIV/NCD multimorbidity and an earlier age of onset of the disease of diabetes and hypertension are observed.[11 12 14 24–27] In response to this, efforts to coordinate NCD programmes with scaled-up NCD medicine supply chains alongside high-quality vertical (stand-alone) HIV programmes have intensified.[28] There is a growing evidence base and shift towards the implementation of integrated NCD/HIV care provision in various SSA countries (eg, in Malawi, South Africa, Botswana, Uganda, Kenya and Tanzania).[22 29–41] Integrated health education and concurrent management of HIV, hypertension and diabetes is being scaled up in vertical and one-stop clinics in the region and can reduce duplication and fragmentation of services; support medicine supply chains; streamline detection and care of comorbidities and multimorbidities; support patient uptake, retention and adherence to treatment; increase viral suppression rates and achieve better control of hyperglycaemia and hypertension; encourage health awareness raising and reduce HIV-related stigma.[21 42–57] See figure 1.

Innovative, more efficient and cost-effective interventions explicitly focused on tackling HIV, hypertension and diabetes comorbidities and multimorbidities in SSA community (particularly rural) settings are required to respond to the increased burden of NCDs, comorbidities and multimorbidities. While community-based or home-based care of HIV using outreach, lay health workers

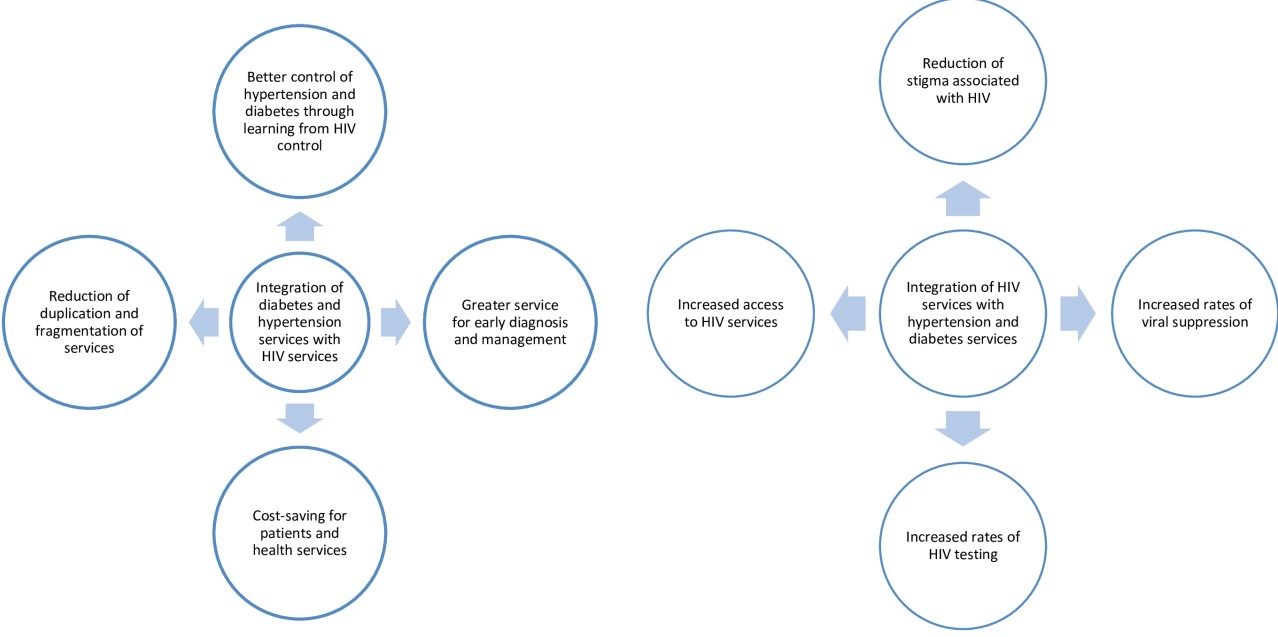

**Benefits for hypertension and diabetes** | **Benefits for HIV control**

**Figure 1** Benefits of integrated HIV, hypertension and diabetes care in general.

and peer support models is expanding in various SSA countries (Uganda, Mozambique and Kenya), community models for diabetes and hypertension care are less developed.[58–62] Our Research Partnership for the Control of Chronic Diseases in Africa (RESPOND-AFRICA) partnership will generate evidence specific to community-based integrated care of HIV, diabetes and hypertension in two SSA countries, which can be used to inform and support decentralisation of integrated care for these conditions. We present here the process evaluation of a pragmatic cluster-randomised trial (INTE-COMM) in Tanzania and Uganda that will compare community-integrated care (HIV, diabetes and hypertension) with standard facility-based integrated care (protocol number ISRCTN15319595).

### The INTE-COMM project

RESPOND-AFRICA has operated in Tanzania and Uganda for many years. We have piloted integrated HIV/NCD care in a feasibility single-arm intervention study in Uganda and Tanzania in the Management of Chronic Conditions in Africa (MOCCA) project (2018–2020).[21 29 38 41] MOCCA was followed by scaled-up integrated care for HIV, diabetes and hypertension in the integrating and decentralising HIV, diabetes and hypertension services in Africa (INTE-AFRICA trial) (2019–2023).[39 40 63 64] INTE-COMM is a 4-year research project funded by the National Institute for Health and Care Research (NIHR) (01/10/2020-30/09/2024). We will closely adhere to the Medical Research Council's (MRC) framework for developing and evaluating complex interventions and frameworks for intervention adaptation and multimorbidity interventions. In a formative phase of 18 months, we have

been guided by community-based models of HIV care that include the provision of medicines, peer support and self-management and a series of collaborative stakeholder consultations with policymakers and patient (service user) groups to design a new community-based integrated model of care for the management of patients with HIV, diabetes and hypertension. This has informed the design of INTE-COMM, which will differ from existing community care approaches by managing multiple conditions (HIV, hypertension and diabetes) in the community via health monitoring and adherence/lifestyle advice (medicines, diets and exercise) provided by community nurses and trained lay workers and devolving NCD drug dispensing to the community level. While INTE-COMM is situated in Tanzania and Uganda, we intend to generate evidence potentially transferable to other community settings in the SSA and wider continent.

### Process evaluation of the INTE-COMM trial

Process evaluations typically evaluate how and whether interventions are delivered as intended and whether such implementation is congruent with the theory underpinning the intervention.[65–69] The MRC guidance for the evaluation of complex health interventions underscores the importance of process evaluation within trials.[66] Building on experiences and lessons learnt in our process evaluation of the INTE-AFRICA trial,[63] the process evaluation of INTE-COMM will focus explicitly on assessing the various system dimensions pertinent to the community care context; for example, how the meanings of HIV, diabetes and hypertension in communities influence receipt of the INTE-COMM intervention; how implementation processes function to facilitate or inhibit access to

care and the mechanisms driving service user/provider and community outcomes.

Using qualitative interviews and non-participant observations, we will document and explore the experiences, attitudes and practices of a wide variety of stakeholders during implementation in order to develop a process-led understanding of how sociostructural factors shape the initiation, delivery and implementation of community-based HIV/NCD service integration over time. We will clarify causal mechanisms and contextual factors associated with variation in outcomes between community-based and facility-based integrated care and assess to what extent resources and activities are supporting the INTE-COMM trial in delivering intended outputs and related clinical and health economic outcomes.

A central focus lies in identifying challenges and contextually relevant strategies or solutions in the community setting that can support and guide the successful implementation and sustainability of community-based integrated prevention, detection and management of HIV with hypertension and diabetes. We are cognizant that this community approach is not devoid of controversy, as service users will be moved away from traditional facility-based integrated care where they are managed by certified clinicians to a community venue where services are delivered by a trained lay worker supported by a nurse. Patients can perceive this as inferior and a far cry from those managed in stand-alone specialist facilities. We also appreciate that HIV is a stigmatising condition, and putting patients together with others without the condition in a community setting may be disconcerting. These potential limitations to the effective delivery of the intervention will be carefully monitored within this process evaluation, and insights will be used to both explain trial findings and inform future wide-scale implementation.

### Theoretical framework: ecological systems theory

The process evaluation will adapt Bronfenbrenner's ecological systems theory[70 71] to conceptualise the INTE-COMM intervention as an event that disrupts complex social systems operating across multiple contextual levels.[67 72] The theory provides an organising structure to facilitate an in-depth understanding of how implementation of the intervention is shaped by the wider context and, importantly for this study, its interaction with the local communities into which INTE-COMM is introduced.[63 67 70–73] Drawing on our previous primary care research in LMIC settings,[73] three elements of context (see figure 2) will be investigated to capture variation in adoption, delivery and maintenance, which are likely to be important factors in outcome differences.

## METHODS AND ANALYSIS
### Study setting

Tanzania and Uganda were chosen as study sites, given their pioneering work in this area.[29 38–40 62 63 74 75] Their public health services are strongly committed to providing services for NCDs and are struggling to scale up provision for diabetes and hypertension in the face of competing health demands, including from HIV, diabetes and hypertension across all socioeconomic strata.[74] See table 1.

### Study population and recruitment

Patients with HIV, diabetes or hypertension who are considered stable on treatment at the health facilities

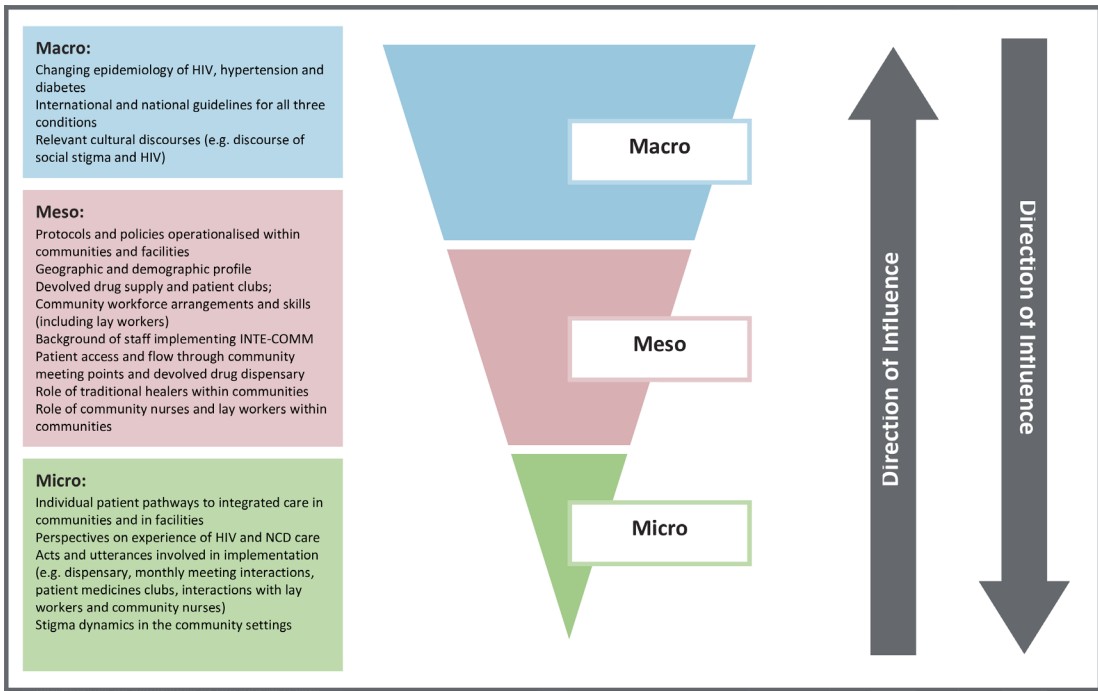

**Figure 2** Potential contextual influences on the INTE-COMM programme implementation cascade.

**Table 1** Profile of Tanzania and Uganda regarding prevalence of three chronic conditions (hypertension, diabetes and HIV)

| Indicators | Uganda | Tanzania |
|---|---|---|
| Income levels | Low | Middle |
| Population size | 49 332 360 | 65 497 748 |
| HIV prevalence | 6.2% | 4.9% |
| Diabetes prevalence | 4.6% | 9.1%, |
| Hypertension prevalence | 33.4% | 26.0% |
| Doctors density/100 000 population | 0.1 | 3 |

will be invited to join the INTE-COMM trial. The trial will evaluate whether community-based integrated care will be effective in improving patient outcomes among PLHIV, diabetes and hypertension as compared with standard facility-based integrated care (including outcomes regarding uptake, retention, acceptability, costs, linkage and potential cost-effectiveness of integrated community care). Patients living with HIV, diabetes or hypertension, stable on treatment at facilities, will be organised into groups of approximately 8–14 persons and randomised to community-based or facility-based integrated care. The study has two co-primary endpoints: a composite endpoint of glycaemia and blood pressure control among individuals with diabetes and/or hypertension and plasma viral load suppression among PLHIV. In the integrated community arm, participants will receive their drugs, adherence support, health education and monitoring at a community venue. Facility-based integrated care will comprise combined health education and clinical management of HIV, diabetes and hypertension. See figure 3.

All trial participants will be followed up for 12 months. A sample size of 116 groups consisting of 12 persons, of which eight with diabetes or hypertension and four with HIV (total of 1392 participants), will provide over 80% power to detect an absolute difference in blood pressure and blood glucose control of 10% at the 5% two-sided significance level. For HIV viral suppression, the trial will have over 80% power to show non-inferiority with a margin of deltaof 8.5%, 7.5% and 5.5%, assuming viral suppression is 85%, 90% and 95%, respectively. To allow for a loss of follow-up of over 10% in the groups, we will recruit 124 groups, comprising 14 persons, giving a target sample of 1736 participants. Each country will enrol a total of 62 groups and 868 participants in the trial (protocol reported elsewhere).

A purposive sampling approach will recruit process evaluation participants. Recruitment will continue until data saturation is reached (see analysis). Service users recruited from those who have already consented to participate in the main trial from both arms, of both genders and on stable treatment for either hypertension, diabetes, HIV or multimorbidity will be recruited

for face-to-face interviews with the help of gatekeepers (facility and community nurses) at three sites (urban, rural and periurban) (estimated n=28 in each country). Replacement of participants (eg, service users) will occur if a significant loss to follow-up or refusal of repeated interviews occurs. Healthcare service providers at both arms (health facility managers, physicians/clinicians, nurses, lay workers, patient club facilitators/peer educators and traditional healers) will also be invited for face-to-face interviews (estimated n=5–10 in each country). Key ministerial policymakers and provincial/regional/district level clinical/health senior management (Director/Programme Manager/Commissioners, Assistant Commissioner of NCDs, Programme Manager/Commissioner of the National AIDS Control Programme and others), non-governmental and international organisations representatives will be identified based on the INTE-AFRICA listing and will be invited for face-to-face or telephonic interviews where it is difficult to meet physically (estimated n=5–10 in each country). Community leaders and members will be identified by gatekeepers in the facility catchment areas and invited to partake in three focus group discussions with 6–10 participants (females, males and community leaders of both genders) with the help of Village Health Team members. Clinical researchers in the INTE-COMM trial will be interviewed in both countries and asked to provide reflections on the implementation process.

## Design of process evaluation

This protocol describes procedures for a process evaluation using qualitative methods within a pragmatic cluster-randomised trial that will compare community-integrated care with facility-based integrated care in Tanzania and Uganda. Qualitative interviews and non-participant observations will be conducted concurrently with the trial and in tandem with the collection of clinical outcomes and health economic data. Qualitative methods will include multistakeholder interviews (patients, community health workers/volunteers, healthcare providers, policymakers, clinical researchers and international and non-governmental organisations) and focus group discussions (community leaders and members), supported by observations of community group meetings in intervention sites. Data collection using these methods will document and explore various dimensions of implementation from diverse perspectives at three timepoints in the trial implementation (at baseline, at 6 months and at 12 months). See table 2 for further detail according to detailed research objectives per participant group.

By using Bronfenbrenner's ecological systems model, this design permits an assessment of the theoretical fidelity of INTE-COMM's implementation, understood for how the intervention is flexibly adapted to function within localised community contexts, rather than a strict assessment of fidelity.[67 68] To do so requires a detailed description of the processes, relationships and contexts involved in the delivery of community-based integrated

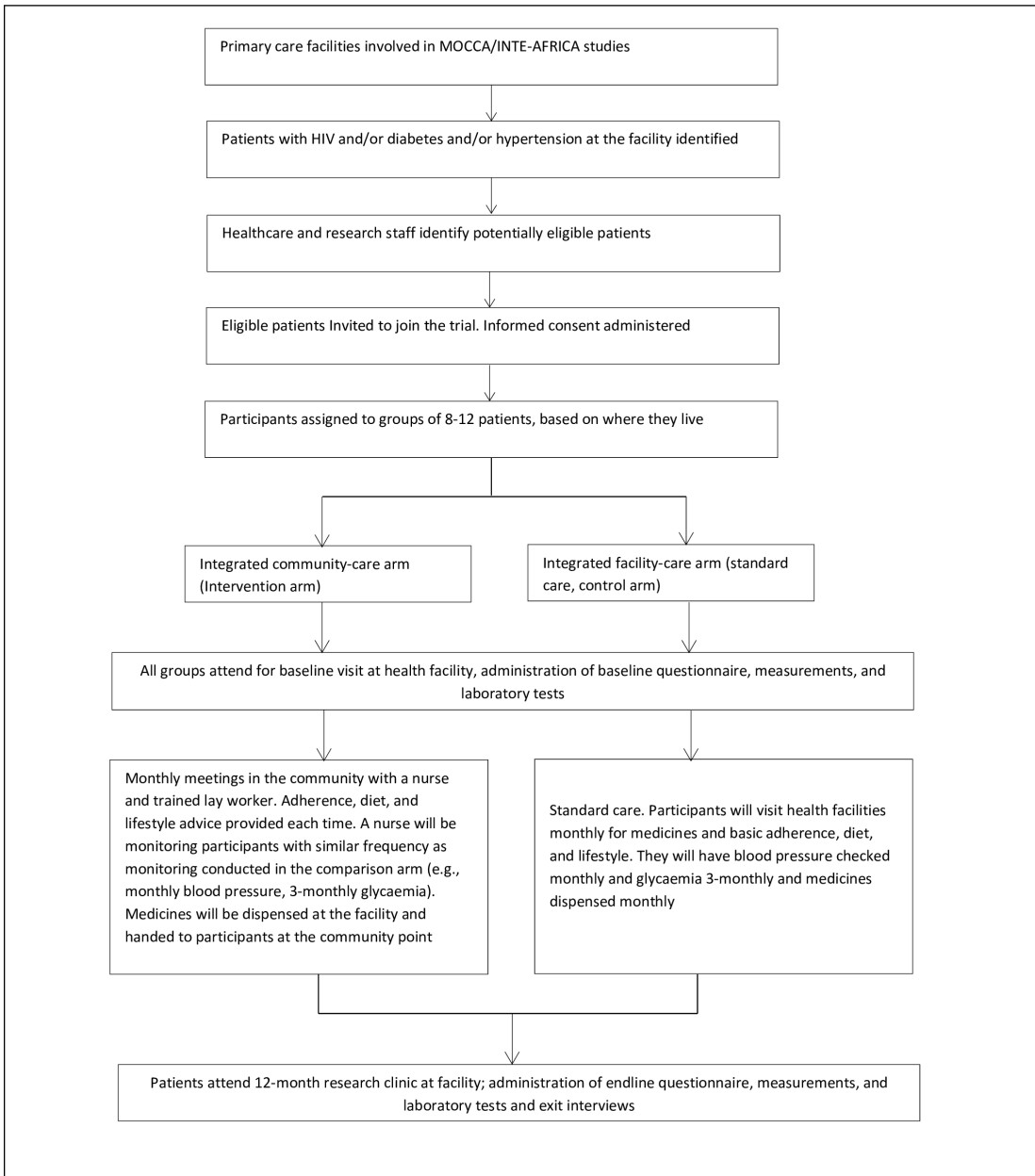

**Figure 3** INTE-COMM study schema.

care and the identification of factors attributing to the failure or success of the programme. It addresses the black box problem in interpreting trial results by improving the understanding of the mechanisms that connect particular intervention components to particular outcomes.[76] This synergistic approach to evaluate, understand and respond will enhance the acceptability of community-level integrated approaches in both countries to support the refinement of care for HIV, diabetes and hypertension, inform health surveillance, support effective drug dispensing, staff capacity building and future policy reforms to involve and educate local communities and sustain and scale up integrated HIV, diabetes and hypertension care in Tanzanian and Ugandan community settings.

**Data collection**

Data collection tools (interview and focus group guides) will explore multistakeholder perceptions and experiences of the decentralised integrated management of HIV, diabetes and hypertension care in community and facility settings. Interview and focus group guides will be designed based on a scoping review of the literature and process evaluation team consultation with the INTE-COMM trial implementers. Questions will be carefully developed in order to develop a phenomenological understanding[77] of the experiences, processes and contextual factors that influence the implementation of the intervention within community ecocultural and cultural systems (see table 2 research objectives and online supplemental file 1). Questions and probes that

**Table 2** Objectives, contextual features and data collection methods of INTE-COMM process evaluation

Objective/research question 1:*What are patients', healthcare providers' and lay health workers' perceptions and experiences of integrated management of HIV, diabetes and hypertension care in the community and at the health facility level?*

| Contextual level/feature | Data collection method | Sources of information |
|---|---|---|
| **Meso:** Time and resources required; utilisation of policies and protocols; organisation and process of monthly meetings; training, organisation and work allocation of nurse and lay health worker; implementation processes and adaptations to embed intervention within communities.<br>**Micro:** Individual patient pathways to integrated care; relationship between intervention and management of hypertension, diabetes and HIV within everyday life. | Interview | Service users (patients stable on treatment) |
| | Interview | Healthcare providers (health facility managers, physicians/clinicians, nurses, lay health workers and patient club facilitators) |
| | Focus groups | Community members or leaders (8–12 participants)per group |
| | Interview | Representatives from patient clubs/groups/peer educators. |
| | Interview | INTE-COMM investigators |

Objective/research question 2:*What key contextual and process factors influence patients' and healthcare providers' uptake and acceptability of integrated management of HIV, diabetes and hypertension care in the community and at the health facility level?*

| | | |
|---|---|---|
| **Macro:** Wider cultural discourses of HIV, hypertension and diabetes (eg, HIV stigma); availability of medicines; socioeconomic, trained staff and structures where services are delivered.<br>**Meso:** Geographic and demographic profile of communities and facilities; medicines supply; workforce arrangements of main social actors involved in intervention delivery (ie, nurses and lay workers, patients, carers and community leaders); local practices and protocols; patient access and flow through community; referral pathways; the role of traditional healers and the different sites and scenes which impact on how INTE-COMM is adapted and delivered (eg, local area management meetings)<br>**Micro-**individual level: influence of privacy and confidentiality in accessing care, trust placed to those delivering the intervention, mobility factor and economic factors. | Interview | Policymakers and Reps of NGOs (ministerial policymakers and provincial/regional/district level clinical/health senior management (Directors/ Programme managers/ Commissioners, Assistant Commissioner of NCDs; Commissioner/Programme manager of the National AIDS Control programme) |
| | Interview | Healthcare providers (health facility managers, physicians/clinicians, nurses, lay health workers, patient club facilitators) peer groups/educators and traditional healers.<br>Service care users (patients stable on treatment) |
| | Focus groups | Community members/leaders (8–12 participants per group) |

Objective/research question 3: *To evaluate the implementation and theoretical fidelity of the intervention*

| | | |
|---|---|---|
| **Meso:** Main (ie, monthly meetings and interactions at dispensaries) activities and subsidiary activities (eg, additional training/meetings and receptionist screening of patients) of intervention delivery and interactional arrangements of each activity: structural arrangement at the point of service delivery (friendly environment, privacy and confidentiality issues). | Non-participant observations with structured field notes | Service delivery processes, the flow of clinical procedures, the nature of services given, listening in patients' and health workers' conversations, freedom in patients entering the service point interactions, discussing and asking questions when waiting for the services |

Objective/research question 4: *What factors can potentially influence the scaling up of integrated management of HIV, diabetes and hypertension at the community and health facility level?*

| Table 2 | Continued | |
|---|---|---|
| **Objective/research question 1:** *What are patients', healthcare providers' and lay health workers' perceptions and experiences of integrated management of HIV, diabetes and hypertension care in the community and at the health facility level?* | | |
| **Micro:** Cultural and structural characteristics, networks and communication and external policies and incentives.<br>**Meso:** Community and facility resource constraints, drug supply and patient medicine clubs and referral pathways. | Interview | Service users |
| | Interview | Healthcare providers (health facility managers, physicians/clinicians, nurses, lay health workers and patient club facilitators) |
| | Interview | Representatives from patient clubs, groups and peer educators |
| | Focus groups | Community leaders and community members (8–12 participants per group) |
| | Interview | Policymakers and reps of NGOs (ministerial policymakers and provincial/regional/district level clinical/health senior management (Directors/ Programme managers/Commissioners, Assistant Commissioner of NCDs; Commissioner/Programme manager of the National AIDS Control programme) |

will unpack and identify various factors that can potentially influence the sustaining and scaling up of integrated management of HIV, diabetes and hypertension in the community and at facility levels will be identified and explored. We include an explicit objective to monitor and identify potential sources of contamination and dilution of the intervention, for example, relating to HIV and whether people in intervention communities are going elsewhere.

Where possible, we will gender-match the interviewer with the participant, particularly in the case of service users. In Tanzania, all interviews and focus groups will be conducted in the Swahili language; in Uganda, interviews with service users and focus groups with community members/leaders of medicine clubs will be conducted in local languages and the remainder will be conducted in English. Data collection tools will therefore be translated accordingly. Audio recordings will be taken with written and verbal consent and transcribed verbatim by the team. Back translation of transcripts from the local language into English will occur for consistency. Non-participant observations of community group meetings will be documented using written field notes to provide a description of the processes followed and the content involved in delivering integrated care.

### Data analysis and synthesis

Iterative sampling and analysis, moving between data collection points and data analysis to test emerging theories, will continue until saturation is reached. This process of analytic reflexivity and triangulation across methods and sources will provide findings to explain the main trial findings and offer clear directions for future efforts to sustain and scale up community-integrated care for HIV, diabetes and hypertension.

Using this reflexive iterative approach, which we have developed in our previous health systems research,[63 73] we will analyse both theoretical fidelity and the interaction between the INTE-COMM intervention, local communities and the wider health system context. This will be achieved by: (1) setting out relevant macro (eg, cultural discourses of NCDs), meso (eg, protocols used in monthly meetings, structural arrangements at service points to observe privacy and confidentiality, lay health worker/nurse arrangements) and micro (eg, talk and behaviour at dispensaries) contextual features relevant to implementation; (2) targeting where likely tensions in implementation are likely to occur at each contextual level; then (3) analysing tensions within targeted activities involved in intervention delivery and (4) considering the consequences of these tensions for how the intervention was implemented and the implications of these for scaled-up implementation. Tensions in implementation are likely to be manifested in training sessions, interactions between lay health workers, peer educators, nurses and members of the community, or through organisational processes for ensuring the intervention is delivered.

To manage the quantity and range of data collected as part of the process evaluation, the iterative analysis will involve working laterally across data types and collection points and situating evidence within the macro, meso and micro contextual frameworks (see table 2). We will seek to provide a broad description of intervention delivery, but instead of allocating equal time to the analysis of each case, we will focus on identifying 'telling cases', triangulating and looking for connections between data.[78] Care will be taken to identify and follow up on deviant cases that do not fit into emerging theories. Emerging theories and the relationship of

the data to the conceptual literature underpinning the intervention will be discussed and refined at team meetings throughout the research. The inductive (iterative) analysis will further unpack how service users, providers, other stakeholders and their communities understand and apply diverse lay health literacies, autonomies and decision-making processes, including healthcare-seeking regarding the three conditions of interest (HIV, diabetes and hypertension) and settings of clinical care.[63 67 76] An electronic data management package (NVivo.14) will support the coding processes (including the pilot coding of a small number of transcripts at the beginning) undertaken by the research team. This process is supported by regular briefing meetings and team discussion around theme allocation, consensus and explanation of potential outliers.

## Credibility and transferability

The process evaluation protocol adheres to recommendations intended to facilitate the standardisation of process evaluation design and reporting.[67] The credibility and reliability of our data will be anchored in the ability to triangulate sociobehavioural and observational data during iterative processes of collection and analysis in order to better understand how different types of evidence enhance the overall interpretation of community-based integrated care. Triangulation will occur across theories, methods (qualitative and observational) between countries (Tanzania and Uganda) and sources of data (narratives, checklists and notes).

The process evaluation also operates concurrently with clinical outcome data and health economics analysis. Taken collectively, process evaluation data combined with clinical and health economics data can inform scalability and transferability to other community settings.

We recognise the potential for selection and information bias as limitations of the trial itself and mitigate them by using a standardised approach to collecting data in an iterative process with continual assessment of information bias and ensuring that research personnel are unaware of participant disease status. Social desirability is addressed in the process evaluation by only providing brief information at the outset of the evaluation in order to avoid priming, using an interview schedule approved by a panel of INTE-COMM experts in terms of sensitivity, conducting qualitative research using skilled local interviewers with limited power relationships between interviewer and participant, using safe and secure settings where the participant feels comfortable or using technology or a telephone, briefing them that there is no right and wrong answer and finally by encouraging them to use anecdotes and experiential evidence to support their views.

## Patient and public involvement

As with all RESPOND-AFRICA projects, proactive community engagement is crucial for the success of INTE-COMM. The INTE-COMM trial is cognizant of the importance of public and patient involvement (PPI) throughout its programme of research and values PPI in enhancing research quality and relevance by providing different perspectives and a sense of ownership. The process evaluation will allow multistakeholder voices to be heard and used. Key community-level stakeholders, such as service users and their families, community members and community care professionals, will be fully involved in guiding the research, acting as research participants and in the implementation of change in community-level integrated health service delivery and integrated care planning. All aspects of the process evaluation are underpinned by participatory action health research and its success and usefulness will be grounded in PPI, participation and engagement in the form of patient/professional identification of research priorities, collaborations and partnerships, expert steering, community participation around health needs and optimal integrated services, awareness-raising activities, the development of print materials, toolkits and training for community nurses, lay members and healthcare professionals.

## Ethics, data management and dissemination

The protocol has been approved by the University College of London (UK), the London School of Hygiene and Tropical Medicine Ethics Committee (UK), the Uganda National Council for Science and Technology and the Uganda Virus Research Institute Research and Ethics Committee (Uganda) and the Medical Research Coordinating Committee of the National Institute for Medical Research (Tanzania). The University College of London, UK, is the trial sponsor. The key ethical principles of voluntary and informed participation, respect for persons, doing no harm, autonomy, privacy, anonymity, confidentiality and safety of participants will be used in all researcher and participant interactions. Written informed consent for participation in interviews and focus groups will be obtained from all participants. All participants will be provided with written information about the research; this will be explained verbally, and they will be informed that their participation is voluntary and that they may withdraw from participation at any time without any penalty.

In terms of data management, the safety of all data will be ensured by (1) applying encrypted password-protected code to all transcripts; (2) storing all data in an encrypted database for up to 10 years and only being accessible by the INTE-COMM research teams in Tanzania, Uganda and the UK; (3) storing all personal identifying information of participants separately from all transcripts and allocating each participant with a unique participant identification number to delink the information from the personal identifiers and (4) destroying all audio recordings on completion of transcription. The transcript files will be uploaded onto the encrypted database, where they will be stored for up to 10 years and thereafter destroyed by the trial team. Across INTE-COMM countries, only anonymised participants' data will be shared during routine updates and briefing sessions. No

personal identifying information (eg, consent forms and audio recordings) will be shared across countries or via any communication means.

Timely communication and publication of findings across a broad range of audiences is essential to driving policy considerations around community-based integrated HIV/NCD care in Africa. Through targeted dissemination, communication, outreach and the provision of innovative tools and models, the project's long-term conceptual, instrumental, capacity building and network influences will centre on achieving an enhanced understanding of the intricacies involved in integrating HIV and NCD services and patient care at the community level. Dissemination and communication will fully and regularly use all established and innovative channels (eg, journal publications, conferences, press releases and webinars) to target all stakeholders, including a variety of audiences and in local languages and English.

**Author affiliations**
[1]Public Health Institute, Liverpool John Moores University, Liverpool, UK
[2]MRC/UVRI and LSHTM Uganda Research Unit, Entebbe, Wakiso, Uganda
[3]Health Systems, Policy and Translational Reseach Section, National Institute for Medical Research, Dar es Salaam, Tanzania, United Republic
[4]Institute for Global Health, University College London, London, UK
[5]School of Life Course and Population Sciences, King's College London, London, UK
[6]Muhimbili Medical Research Centre, National Institute for Medical Research Tanzania, Dar es Salaam, Tanzania, United Republic of

**Contributors** All authors contributed to the conceptualisation of the research and contributed to writing the manuscript. MCVH, MA, EHS, MN, IN, JM, AK, SGM and SJ designed the process evaluation protocol. SJ, JO, FXK and JB led the development of the INTE-COMM trial. MCVH drafted the manuscript and all co-authors edited and commented on subsequent drafts. All authors approved the final draft for submission. All authors agree to be accountable for all aspects of the work in ensuring that questions related to the accuracy or integrity of any part of the work are appropriately investigated and resolved.

**Funding** This work was supported by NIHR; GHPSR Project: NIHR 131273.

**Competing interests** None declared.

**Patient and public involvement** Patients and/or the public were involved in the design, or conduct, or reporting, or dissemination plans of this research. Refer to the Methods section for further details.

**Patient consent for publication** Not applicable.

**Ethics approval** This study involves human participants and was approved by the University College of London (UK), the London School of Hygiene and Tropical Medicine Ethics Committee (UK) (ref 281-221) and the Uganda National Council for Science and Technology and Uganda Virus Research Institute Research and Ethics Committee (Uganda) (ref GC 127/872) and the Medical Research Coordinating Committee of the National Institute for Medical Research (Tanzania). Participants gave informed consent to participate in the study before taking part.

**Provenance and peer review** Not commissioned; externally peer reviewed.

**Data availability statement** No data are available. Not applicable.

**ORCID iDs**
Marie-Claire Van Hout http://orcid.org/0000-0002-0018-4060
Elizabeth Henry Shayo http://orcid.org/0000-0001-9131-1784
Francis Xavier Kasujja http://orcid.org/0000-0002-4081-2826
Sayoki Godfrey Mfinanga http://orcid.org/0000-0001-9067-2684

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
