## [Reviewer comments · BMJ Open]

ARTICLE DETAILS

TITLE (PROVISIONAL)	Decentralizing chronic disease management in sub-Saharan Africa: a protocol for the qualitative process evaluation of community-based integrated management of HIV, diabetes, and hypertension in Tanzania and Uganda.
AUTHORS	Van Hout, Marie-Claire; Akugizibwe, Mathias; Shayo, Elizabeth; Namulundu, Moreen; Kasujja, Francis; Namakoola, Ivan; Birungi, Josephine; Okebe, Joseph; Murdoch, Jamie; Mfinanga, Sayoki; Jaffar, Shabbar

VERSION 1 – REVIEW

REVIEWER	Nicol, Jeannine Stellenbosch University, Global Health
REVIEW RETURNED	17-Oct-2023

GENERAL COMMENTS	Abstract: the abstract is well written however there is less information on the description of the methods in terms of design given that qualitative piece is part of the pragmatic trial. There is no description on data analysis. Main text: Line 5-8 (pg 6): "To the best of our knowledge there has been no clear evidence to date for a community-based model of integrated HIV& NCD care in SSA." I think this statement is somewhat not true as there have been number of studies conducted in SSA looking at integrated care services for communicable and non-communicable diseases including community-based interventions. I would suggest the authors to refer to a recent scoping review on Integrated models for NCDs in LMICs. in light of existing studies in the literature on integrated models of care for multimorbidity in LMICs, what is new that / innovation that this pragmatic trail is adding to existing literature or body of knowledge? WHO has recently published a guideline on management of multimorbidity focusing both NCDs and CDs that I would recommend the aithors to review so thatthey are inspired on interventions that have proven to be working thus far. Below are some of useful references that to look at that could assist the uathors frmaining their research interventions. Rohwer, A., Toews, I., Uwimana-Nicol, J. et al. Models of integrated care for multi-morbidity assessed in systematic reviews: a scoping review. BMC Health Serv Res 23, 894 (2023). https://doi.org/10.1186/s12913-023-09894-7 Uwimana J, Zarowsky C, Hausler H, Swanevelder S, Tabana H, Jackson D. Community-based intervention to enhance provision of integrated TB-HIV and PMTCT services in South Africa. Int J Tuberc Lung Dis. 2013 Oct;17(10 Suppl 1):48-55. doi: 10.5588/ijtld.13.0173. PMID: 24020602.
--

	Rohwer A, Uwimana Nicol J, Toews I, et al Effects of integrated models of care for diabetes and hypertension in low-income and middle-income countries: a systematic review and meta-analysis BMJ Open 2021;11:e043705. doi: 10.1136/bmjopen-2020-043705 Methods: Lines 33-54- the qualitative process evaluation is not a research design rather a type of evaluation. The authors should consider describing the actual design for the qualitative research methods to be used The process evaluation using qualitative methods is within a trial. will the qualitative piece take place after the trial or concurrently? this needs to be clarified. study participants recruitment - how that be conducted especially for the beneficiaries of the study? what will be the sample size and sampling methods to be used? It will be useful to have a clear objective or theme/topic that each category of the study participant will focusing on to collect data. Data analysis plan is not that much details given the diversity of participants involved representing different roles, power and perspectives. A description of the ethnographic approach needs to be well defined to determine what data will be using such approach given the multi-methods approach used in data collection of which observations are part of. Will there be any level of triangulation of data during data analysis or data will be analyzed separately. if so how? Data management and sharing: There is no description on how the data will be managed and shared among countries and institutions involved.
--	---

REVIEWER	Brkic, Alen Sørlandet Sykehus HF, Research Department
REVIEW RETURNED	21-Dec-2023

GENERAL COMMENTS	To Editor: Due to time constraints, I could not thoroughly validate and review all the citations in the manuscript. However, based on my examination of the first reference and its associated content, there seems to be a discrepancy between the original source and how it has been paraphrased or cited in the paper. While I highlight only one instance here, I have noticed a few others that raise some concerns. This is only in the very first aspect of the paper. I strongly recommend a comprehensive review of all sentences exceeding 30 words. Such sentences, while sometimes necessary, can often impede readability. It would be beneficial for the clarity and overall flow of the manuscript if these longer sentences were either shortened or divided into multiple sentences. Regarding the manuscript's layout, I noticed an issue with the alignment of the line numbers. They do not correspond accurately
--

with the actual sentences in the text. Due to this, pinpointing specific locations for reference becomes somewhat challenging. I will describe the places of my comments to the best of my ability.

Thank you for the opportunity to review your paper.

The Manuscript:

This well-described and important protocol will have great potential for further studies and other protocols in a similar field. Most of my comments are focused on the first aspect of the manuscript. In my opinion, the method section is significantly better than the first part of the manuscript regarding readability and clarity (except for some very long sentences). I was also picky about the graphical elements, but in my opinion, clear visual illustrations are important to any scientific paper attempting to disseminate information.

Regarding the first sentence on page 4 in the Introduction:
The sentence aims to convey the significance of the increase in NCD deaths, yet the numerical data provided does not accurately reflect this. I recommend using specific numbers to illustrate the rise in NCD deaths for a more impactful presentation. For instance, according to the "Global Burden of Disease Study 2019" by the Institute for Health Metrics and Evaluation (IHME) [<https://vizhub.healthdata.org/gbd-results>], the percentage of deaths due to NCDs increased from 60.45% in 1999 to 74.37% in 2019, which is approximately a 15% rise over 20 years. This kind of precise data would significantly bolster the statement's effectiveness.
The manuscript refers to "the most recent data," but it's important to note that this data, sourced from the GBD database, only extends up to 2019. As we are now five years beyond this point, the data may not fully represent the current situation. Perhaps it would be better to present the year being assessed.

Regarding the third sentence on page 4 of the Introduction, there is an inconsistency in the terminology used by the authors. Initially, the term NCD death is used, but it shifts to NCD related death. Please maintain consistency in terminology throughout the manuscript. Also, the term NCD related death potentially encompasses a broader range of causes than NCD death. For instance, cancer death typically implies death where cancer is the primary cause. In contrast, cancer related death might include deaths due to complications from cancer treatments or operations. In the last sentence of the first paragraph on page 4 of the Introduction, there appears to be a notable similarity with a sentence in the first reference cited. The manuscript states, "the leading metabolic risk factor globally is elevated blood pressure (to which 19% of global deaths are attributed) (1), followed by raised

blood glucose and overweight and obesity." This phrasing closely mirrors the language used in the reference.

In the first sentence of the second paragraph on page 4, Introduction, the phrase "reduce reducing" seems to be an error.

In the second paragraph of page 4, Introduction, there is a sentence stating, "86% of premature deaths and 85% of deaths in people aged 30-69 years related to NCDs occur in LMICs.": Upon reviewing the cited source, I could not locate the specific figure of "85%" as mentioned in the manuscript.

Please clarify whether these two categories (premature deaths with deaths in people aged 30-69 years) are synonymous or not within the context of the manuscript.

The term "vertical HIV programming", while specific to public health programming, may not be universally understood by all readers, especially those not familiar with public health terms. Consider explaining it differently or providing a definition.

The last sentence of the largest paragraph on page 5, Introduction. This sentence is approximately 100 words in length and presents significant readability challenges. I strongly suggest improving the readability of this sentence.

I have observed a minor misalignment of the rectangles in Figure 1 in the manuscript.

I find the sentence between pages 5 and 6 somewhat challenging to read. Could it be improved?

In Figure 2 of the manuscript, I believe several improvements are needed to enhance its visual clarity.

The rectangles in the figure should be appropriately aligned and should be of equal size.

The text within each rectangle should be centralized along the y-axis.

Consistent spacing between the rectangles

All arrows should be of equal length

The abbreviation "BP" used in the figure should either be defined in the legend or text of the figure, or it should be written out in full.

The sentence between pages 7 and 8 is hard to read and is overly long. Please consider dividing it into two.

I suggest some changes in Figure 3 to enhance its visual presentation and clarity:

Place the triangle representing Macro, Meso, and Micro levels on the leftmost side of the frame, aligned with the square that contains the text. Ensure that the triangle remains within the frame's boundaries.

Instead of having a separate frame for the labels Macro, Meso, and Micro, incorporate these labels directly within the respective sections of the triangle.

Position the descriptive text next to the triangle on the right side.

Remove any titles from this section to maintain simplicity and focus on the content.

If the text corresponding to Macro, Meso, and Micro sections is too close, consider adding small, thin lines between these sections for better visual separation and organization.

Utilize bullet points for the text, ensuring there is a slight separation between each point for clarity.

The 'Direction of Influence' label currently appears to be overly prominent. Consider reducing its size, especially if additional space is needed for the text.

Table 1 is called Table 2. The manuscript has two Tables called Table 2.

	I have a few suggestions for Table 1, titled "Profile of Tanzania and Uganda," What is gained by filling the table with full references instead of the citation itself? Maxed the reference amount? If not, I think it's better not to have the full reference in the table. This saves space and improves clarity. There is a surplus parenthesis in the cell for Uganda/Doctor Density. Ensure uniformity in the presentation of numerical data by keeping the same number of decimal places for all figures. For instance, even if the number is a whole number, like 26 or 3, it should be presented as 26.0 or 3.0 to match the format of other numbers in the table. There seems to be inconsistency in the usage of phrase X base Y (e.g., clinical base settings), which can lead to confusion or misinterpretation. Do you mean clinical-based settings? If possible, I would try to limit the second table to one full page – instead of having it split.
--	---

VERSION 1 – AUTHOR RESPONSE

Reviewer: 1

Dr. Jeannine Nicol, Stellenbosch University, University of Rwanda College of Medicine and Health Sciences

Comments to the Author:

Abstract: the abstract is well written however there is less information on the description of the methods in terms of design given that qualitative piece is part of the pragmatic trial. There is no description on data analysis.

AUTHOR RESPONSE: Thank you this is now included, and clearly states our iterative framework approach to data collection and analysis. Iterative sampling, triangulation across perspectives and analysis moving between data collection points and data analysis to test emerging theories will continue under saturation is reached..

Main text:

Line 5-8 (pg 6): "To the best of our knowledge there has been no clear evidence to date for a community-based model of integrated HIV& NCD care in SSA." I think this statement is somewhat not true as there have been number of studies conducted in SSA looking at integrated care services for communicable and non-communicable diseases including community-based interventions. I would suggest the authors to refer to a recent scoping review on Integrated models for NCDs in LMICs. in light of existing studies in the literature on integrated models of care for multimorbidity in LMICs, what is new that / innovation that this pragmatic trail is adding to existing literature or body of knowledge? WHO has recently published a guideline on management of multimorbidity focusing both NCDs and CDs that I would recommend the authors that they are inspired on interventions that have proven to be working thus far. Below are some of useful references that to look at that could assist the authors remaining their research interventions.

Rohwer, A., Toews, I., Uwimana-Nicol, J. et al. Models of integrated care for multi-morbidity assessed in systematic reviews: a scoping review. BMC Health Serv Res 23, 894

(2023). <https://doi.org/10.1186/s12913-023-09894-7>

Uwimana J, Zarowsky C, Hausler H, Swanevelder S, Tabana H, Jackson D. Community-based intervention to enhance provision of integrated TB-HIV and PMTCT services in South Africa. Int J Tuberc Lung Dis. 2013 Oct;17(10 Suppl 1):48-55. doi: 10.5588/ijtld.13.0173. PMID: 24020602.

Rohwer A, Uwimana Nicol J, Toews I, et al Effects of integrated models of care for diabetes and hypertension in low-income and middle-income countries: a systematic review and meta-analysis *BMJ Open* 2021;11:e043705. doi: 10.1136/bmjopen-2020-043705

AUTHOR RESPONSE: Thank you for the information. We apologise for the lack of clarity in our original drafting. We recognise the development of integrated care in the sub-Saharan African region with regard to HIV, TB, cancers, women's health and so forth. In response to your query, we have now carefully revised those sections to be clear in its focus in on the two countries of focus and the lack of evidence around HIV, diabetes and hypertension specific integrated care models in the community in those countries. This provides a clear rationale for the trial.

Methods: Lines 33-54- the qualitative process evaluation is not a research design rather a type of evaluation. The authors should consider describing the actual design for the qualitative research methods to be used The process evaluation using qualitative methods is within a trial. will the qualitative piece take place after the trial or concurrently? this needs to be clarified.

AUTHOR RESPONSE: Thank you very much, we have clarified the design and methods sections. Iterative data collection and analysis is used to evaluate the trial, with data collection points at the baseline, mid and at 12 months. This is clearly outlined with a section describing the trial methodology and recruitment, and the process evaluation design, process and analysis.

study participants recruitment - how that be conducted especially for the beneficiaries of the study? what will be the sample size and sampling methods to be used?

AUTHOR RESPONSE: Thank you very much, this detail is provided with regard to the trial and the process evaluation.

It will be useful to have a clear objective or theme/topic that each category of the study participant will focusing on to collect data.

AUTHOR RESPONSE: Thank you very much, this detail is provided in the text and also in the large Table 2.

Data analysis plan is not that much details given the diversity of participants involved representing different roles, power and perspectives. A description of the ethnographic approach needs to be well defined to determine what data will be using such approach given the multi-methods approach used in data collection of which observations are part of. Will there be any level of triangulation of data during data analysis or data will be analyzed separately. if so how?

AUTHOR RESPONSE: Thank you very much, this detail is provided in the section on data analysis and triangulation is outlined in the subsequent section 'Credibility and transferability'.

Data management and sharing: There is no description on how the data will be managed and shared among countries and institutions involved.

AUTHOR RESPONSE: Thank you very much, this detail is provided in the section on ethics, data management and dissemination.

Reviewer: 2

Dr. Alen Brkic, Sørlandet Sykehus HF

Comments to the Author:

See section of attached document.

To Editor: Due to time constraints, I could not thoroughly validate and review all the citations in the manuscript. However, based on my examination of the first reference and its associated content, there seems to be a discrepancy between the original source and how it has been paraphrased or cited in the paper. While I highlight only one instance here, I have noticed a few others that raise some concerns. This is only in the very first aspect of the paper.

AUTHOR RESPONSE: Thank you very much, the first few paragraphs have been extensively overhauled and checked for accuracy.

I strongly recommend a comprehensive review of all sentences exceeding 30 words. Such sentences, while sometimes necessary, can often impede readability. It would be beneficial for the clarity and overall flow of the manuscript if these longer sentences were either shortened or divided into multiple sentences.

AUTHOR RESPONSE: Thank you very much, the manuscript has had a heavy copy edit for brevity.

Regarding the manuscript's layout, I noticed an issue with the alignment of the line numbers. They do not correspond accurately with the actual sentences in the text. Due to this, pinpointing specific locations for reference becomes somewhat challenging. I will describe the places of my comments to the best of my ability.

AUTHOR RESPONSE: Thank you very much, this is applied by BMJ when generating the pdf and beyond our control.

Thank you for the opportunity to review your paper.

AUTHOR RESPONSE: Thank you very much for your time and very helpful comments.

The Manuscript:

This well-described and important protocol will have great potential for further studies and other protocols in a similar field. Most of my comments are focused on the first aspect of the manuscript. In my opinion, the method section is significantly better than the first part of the manuscript regarding readability and clarity (except for some very long sentences). I was also picky about the graphical elements, but in my opinion, clear visual illustrations are important to any scientific paper attempting to disseminate information.

AUTHOR RESPONSE: Thank you very much, we have revised the introduction.

1. Regarding the first sentence on page 4 in the Introduction:

- The sentence aims to convey the significance of the increase in NCD deaths, yet the numerical data provided does not accurately reflect this. I recommend using specific numbers to illustrate the rise in NCD deaths for a more impactful presentation. For instance, according to the "Global Burden of Disease Study 2019" by the Institute for Health Metrics and Evaluation (IHME) [<https://vizhub.healthdata.org/gbd-results>], the percentage of deaths due to NCDs increased from 60.45% in 1999 to 74.37% in 2019, which is approximately a 15% rise over 20 years. This kind of precise data would significantly bolster the statement's effectiveness.
- The manuscript refers to "the most recent data," but it's important to note that this data, sourced from the GBD database, only extends up to 2019. As we are now five years beyond this point, the data may not fully represent the *current situation*. Perhaps it would be better to present the year being assessed.

AUTHOR RESPONSE: Thank you very much, this section has been checked and revised.

2. Regarding the third sentence on page 4 of the Introduction, there is an inconsistency in the terminology used by the authors. Initially, the term NCD death is used, but it shifts to NCD related death. Please maintain consistency in terminology throughout the manuscript. Also, the term NCD related death potentially encompasses a broader range of causes than NCD death. For instance, cancer death typically implies death where cancer is the primary cause. In contrast, cancer related death might include deaths due to complications from cancer treatments or operations.

AUTHOR RESPONSE: Thank you very much, this section has been checked and revised.

3. In the last sentence of the first paragraph on page 4 of the Introduction, there appears to be a notable similarity with a sentence in the first reference cited. The manuscript states, "the leading metabolic risk factor globally is elevated blood pressure (to which 19% of global deaths are attributed) (1), followed by raised blood glucose and overweight and obesity." This

phrasing closely mirrors the language used in the reference.

AUTHOR RESPONSE: Thank you very much, this section has been checked and revised.

4. In the first sentence of the second paragraph on page 4, Introduction, the phrase "reduce reducing" seems to be an error.

AUTHOR RESPONSE: Thank you very much, this has been removed.

5. In the second paragraph of page 4, Introduction, there is a sentence stating, "86% of premature deaths and 85% of deaths in people aged 30-69 years related to NCDs occur in LMICs.":

- Upon reviewing the cited source, I could not locate the specific figure of "85%" as mentioned in the manuscript.

AUTHOR RESPONSE: Thank you very much, this was cited by the Global Alliance of Chronic Diseases.

- Please clarify whether these two categories (*premature deaths* with *deaths in people aged 30-69 years*) are synonymous or not within the context of the manuscript.

AUTHOR RESPONSE: Thank you very much, these are the same and we use 'aged under 70 years in line with WHO.

6. The term "vertical HIV programming", while specific to public health programming, may not be universally understood by all readers, especially those not familiar with public health terms. Consider explaining it differently or providing a definition.

AUTHOR RESPONSE: Thank you very much, we have included standalone in brackets the first time vertical is mentioned. Kindly see

www.ncbi.nlm.nih.gov/pmc/articles/PMC2148229/

[Understanding the persistence of vertical \(stand-alone\) HIV clinics in the health system in Uganda: a qualitative synthesis of patient and provider perspectives | BMC Health Services Research | Full Text \(biomedcentral.com\)](http://www.ncbi.nlm.nih.gov/pmc/articles/PMC2148229/)

7. The last sentence of the largest paragraph on page 5, Introduction. This sentence is approximately 100 words in length and presents significant readability challenges. I strongly suggest improving the readability of this sentence.

AUTHOR RESPONSE: Thank you very much, this is edited.

8. I have observed a minor misalignment of the rectangles in Figure 1 in the manuscript.

AUTHOR RESPONSE: Thank you very much, this is edited.

9. I find the sentence between pages 5 and 6 somewhat challenging to read. Could it be improved?

AUTHOR RESPONSE: Thank you very much, this is edited.

10. In Figure 2 of the manuscript, I believe several improvements are needed to enhance its visual clarity.

- The rectangles in the figure should be appropriately aligned and should be of equal size.
- The text within each rectangle should be centralized along the y-axis.
- Consistent spacing between the rectangles
- All arrows should be of equal length
- The abbreviation "BP" used in the figure should either be defined in the legend or text of the figure, or it should be written out in full.

AUTHOR RESPONSE: Thank you very much, this is edited.

11. The sentence between pages 7 and 8 is hard to read and is overly long. Please consider dividing it into two.

AUTHOR RESPONSE: Thank you very much, this is edited.

12. I suggest some changes in Figure 3 to enhance its visual presentation and clarity:

- Place the triangle representing Macro, Meso, and Micro levels on the leftmost side of the frame, aligned with the square that contains the text. Ensure that the triangle remains within the frame's boundaries.
- Instead of having a separate frame for the labels Macro, Meso, and Micro, incorporate these labels directly within the respective sections of the triangle.
- Position the descriptive text next to the triangle on the right side. Remove any titles from this section to maintain simplicity and focus on the content.
- If the text corresponding to Macro, Meso, and Micro sections is too close, consider adding small, thin lines between these sections for better visual separation and organization.
- Utilize bullet points for the text, ensuring there is a slight separation between each point for clarity.
- The 'Direction of Influence' label currently appears to be overly prominent. Consider reducing its size, especially if additional space is needed for the text.

AUTHOR RESPONSE: We have already published different iterations of this figure in its current form in five previous articles on health systems strengthening research in LMICS (van Hout et al., 2023; van Rensburg et al., 2022; Murdoch et al., 2021; Murdoch et al., 2020; Murdoch et al., 2018). It is important that we maintain a coherent narrative with this previous work and are therefore reluctant to make these changes without a clear illustration of how this alternative formatting would provide a clear improvement.

13. Table 1 is called Table 2. The manuscript has two Tables called Table 2.

AUTHOR RESPONSE: Thank you very much, we could not find this error but we have checked all numbers are correct.

14. I have a few suggestions for Table 1, titled "Profile of Tanzania and Uganda,"

- What is gained by filling the table with full references instead of the citation itself? Maxed the reference amount? If not, I think it's better not to have the full reference in the table. This saves space and improves clarity.
- There is a surplus parenthesis in the cell for Uganda/Doctor Density.
- Ensure uniformity in the presentation of numerical data by keeping the same number of decimal places for all figures. For instance, even if the number is a whole number, like 26 or 3, it should be presented as 26.0 or 3.0 to match the format of other numbers in the table.

AUTHOR RESPONSE: Thank you very much, this is edited and the reference column is removed.

15. There seems to be inconsistency in the usage of phrase X base Y (e.g., clinical base settings), which can lead to confusion or misinterpretation. Do you mean clinical-based settings?

AUTHOR RESPONSE: Thank you very much, this is edited throughout and refers to integrated community-based or integrated facility-based consistently throughout.

16. If possible, I would try to limit the second table to one full page – instead of having it split

*Please see attached document from this reviewer

AUTHOR RESPONSE: Thank you very much, this is edited down and is one page.

Reviewer: 1

Competing interests of Reviewer: none

Reviewer: 2

Competing interests of Reviewer: No competing interests

Kind Regards

Professor [name redacted for anonymity] corresponding author.

VERSION 2 – REVIEW

REVIEWER	Brkic, Alen Sørlandet Sykehus HF, Research Department
REVIEW RETURNED	18-Feb-2024
GENERAL COMMENTS	All comments are well described and adjusted for. No further comments.